# Peripheral Interventional Strategy Assessment (PISA) for Diabetic Foot Ulcer Revascularization: Preliminary Outcomes of a Multidisciplinary Pilot Study

**DOI:** 10.3390/diagnostics13182879

**Published:** 2023-09-08

**Authors:** Raffaella Berchiolli, Giulia Bertagna, Daniele Adami, Alberto Piaggesi, Elisabetta Iacopi, Francesco Giangreco, Lorenzo Torri, Nicola Troisi

**Affiliations:** 1Vascular Surgery Unit, Department of Translational Research and New Technologies in Medicine and Surgery, University of Pisa, 56126 Pisa, Italy; raffaella.berchiolli@unipi.it (R.B.); giuliaberty.it@hotmail.it (G.B.); danieleadami71@gmail.com (D.A.); lorenzo.torri@gmail.com (L.T.); 2Diabetic Foot Section, Department of Medicine, University of Pisa, 56126 Pisa, Italy; alberto.piaggesi@med.unipi.it (A.P.); elisabettaiacopi@gmail.com (E.I.); francescogiangreco@gmail.com (F.G.)

**Keywords:** chronic limb-threatening ischemia, duplex ultrasonography, lower limb revascularization

## Abstract

Background: Digital subtraction angiography (DSA) still represents the gold standard for anatomical arterial mapping and revascularization decision-making in patients with chronic limb-threatening ischemia (CLTI), although DUS (Doppler Ultrasound) remains a primary non-invasive examination tool. The Global Vascular Guidelines established the importance of preoperative arterial mapping to guarantee an adequate in-line flow to the foot. The aim of this study was to evaluate the accuracy of DUS in guiding therapeutic vascular treatments on the basis of Global Vascular Guidelines without the need of a second-level examination. Methods: Between January 2022 and June 2022, all consecutive patients with CLTI to be revascularized underwent clinical examination and DUS without further diagnostic examinations. Primary outcomes assessed were technical success, and 30-day mortality. Secondary outcomes were 1-year amputation free survival, and time between evaluation and revascularization. Results: Sixty-eight patients with a mean age of 73.6 ± 8.5 years underwent lower limb revascularization. Technical success was 100%, and the 30-day mortality rate was 2.9%. Mean time between evaluation and revascularization was 29 ± 17 days. One-year amputation free survival was 97.1%. Conclusions: DUS without further diagnostic examinations can accurately assess the status of the vascular tree and foot runoff, providing enough information about target vessels to guide revascularization strategies.

## 1. Introduction

Chronic limb-threatening ischemia (CLTI) still represents a life-threatening condition with a high risk of major adverse cardiovascular events (MACEs) and major adverse limb events (MALEs), as well as one of the highest impact burdens on the economic healthcare system [1,2,3]. Moreover, the Global Vascular Guidelines (GVGs) underline the non-negligible 1-year mortality, limb loss, and a decreased quality of life in CLTI patients [4]. Therefore, revascularization—either surgical or endovascular—represents the cornerstone of treatment. Indeed, it is performed to provide sufficient blood flow to the extremities in at least 50% and up to 90% of CLTI patients in certain tertiary referral centers, whereas minor and major amputations are carried out when all previous treatments cannot be used or have failed.

Concerning diabetic patients, indications for revascularization are more complex, taking into account foot perfusion, wounds’ degree, and infections [5], assessed using Toe Pressure (TP), Ankle Pressure (AP), Ankle Brachial Index (ABI), and Transcutaneous Oxygen Pressure (TcPO2) as first line diagnostic methods [6,7]. Regardless, second-level vascular imaging is still mandatory to confirm diagnosis, indications, and to plan for revascularization strategies.

Digital subtraction angiography (DSA) still represents the gold standard for the evaluation of inflow and outflow arteries prior to revascularization [8,9], although it is burdened by local and systemic complications. In this context, Doppler Ultrasound (DUS) has emerged as a well-recognized primary non-invasive tool for patients with PAD. In fact, DUS is harmless, well-tolerated by patients, significantly less expensive than DSA, and provides anatomic and hemodynamic information [10,11,12].

Furthermore, time to revascularization is essential to achieve a successful outcome and to reduce tissue loss, as described by the statement “time is tissue” [13]. Second-level investigations may enlarge the time between diagnosis and treatment, so that several studies have already investigated the reliability of DUS for femoropopliteal and infrapopliteal lesions imaging [14,15,16]. However, the high operator dependency of this method and the subsequent interpretation continues to be an issue, as well as the need for a roadmap-like image of the entire vascular tree. This is the reason why a lot of surgeons continue to rely on traditional DSA for surgical planning.

The aims of this study were to evaluate the accuracy of isolated DUS in guiding therapeutic vascular treatments compared to that of standard second-level imaging, and to analyze our results of consecutive revascularizations based on Duplex Ultrasound Arterial Mapping (DUAM) as a sole diagnostic method without other second-level examinations.

## 2. Materials and Methods

### 2.1. Patients’ Selection

In the period between January 2022 and June 2022, 128 consecutive patients with DFUs were evaluated in a dedicated outpatient setting, in our tertiary referral center. Patients with high suspicion of CLTI were directly selected by a diabetic foot specialist, based on the following criteria: positive history of atherosclerotic disease; typical clinical symptoms; significative noninvasive tests, such as ABI score < 0.5, TBI < 0.7, TcPO2 < 50 mmHg; and/or previous lower limb revascularizations. All patients selected underwent a second complete evaluation by a multidisciplinary team, including diabetic foot specialists, podiatrists, and vascular surgeons with great expertise in PAD treatment (Figure 1).

### 2.2. Initial Clinical Evaluation

Clinical examinations were performed in a single (“one stop”) visit, where indications for revascularizations and treatments’ modalities were assessed by the multidisciplinary team. The evaluation started with a physical examination, wound inspection, and measurements of AP, ABI, and TBI when necessary, in order to classify patients according to the Rutherford classification, but mostly to the Wound, Ischemia, and foot Infection (WIfI) classification [5]. The latter was primarily assessed to estimate a 1-year amputation risk and the potential benefits from revascularization. Once patients were categorized on a clinical basis, they underwent a complete DUAM to evaluate the arterial tree’s status.

### 2.3. Duplex Ultrasonography

All ultrasonographic examinations, as well as all revascularizations, were performed by the same vascular surgeon, who is one of the authors (D.A.), in order to guarantee the reproducibility of both scanning and outcomes. Specifically, he has a twenty-five-year experience in the management of CLTI patients, and a caseload of approximately one thousand DUAM performed. Despite the fact that the learning curve to perform a high-quality DUS may be long, the standardized approach used in our unit allows surgeons to achieve the required skills in a shorter period.

The next-generation MyLab™X75 machine (Esaote^TM^, Genova, Italy) was used to carry out all arterial mapping. A variety of probes (linear 7–4 MHz, linear 10–5 MHz, linear 12–5 MHz, and curvilinear 2–5 MHz) were used during the same examination to obtain the highest quality of B-mode, color-flow images, and Doppler velocity spectra. Hemodynamically significant stenosis ≥ 50% and ≥70% were defined with a peak systolic velocity (PSV) ratio ≥ 2 and ≥3, respectively. The arterial tree was screened from the distal abdominal aorta just below renal arteries to pedal arteries, assessing the length and the degree of focal arterial stenoses.

### 2.4. Arterial Mapping Protocol

The examination started with patients in a supine position, with mild genuflection and the abduction of the index limb. Initially, femoral bifurcation was assessed. Then, popliteal artery was scanned in both medial and posterior approaches, to better evaluate it in the entire length. The tibio-peroneal trunk, posterior tibial, and plantar arteries were assessed via a medial approach, while anterior tibial and dorsalis pedis arteries through a lateral one. Then, the infrarenal aorta and ipsilateral and contralateral iliac axes were studied, regardless of the presence of disease. At the end of the arterial evaluation, a complete scan of the vein heritage was performed, including great saphenous vein (GSV), small saphenous vein (SSV), and superficial arm veins.

Second-level preoperative imaging, such as computed tomography angiography (CTA), magnetic resonance imaging (MRI), or DSA, was only performed when ultrasound images of the essential arterial segments to plan revascularization were either incomplete or uncertain, or in absence of distal runoff by DUS. The majority of patients (16/20, 80.0%) underwent a preoperative CTA: of these, 11 patients (68.7%) for investigating the aorto-iliac segment, 4 patients (25.0%) for assessing the tibial vessels, and only one patient (6.3%) for evaluating correctly the degree of superficial femoral artery (SFA) stenosis. Then, 3/20 (15.0%) patients received a DSA: 2 of these (66.7%) to investigate the infrapopliteal segment and only one (33.3%) to assess the iliac axes. Only one patient (1/20, 5.0%) underwent an MRI to evaluate the target runoff vessel prior to surgical revascularization.

### 2.5. Final Assessment and Decision Making

All the aforementioned criteria about clinical and DUS evaluations were used to draw a scheme of the arterial mapping of the index limb, which outlined areas of significant segmental or sequential stenoses and occlusions, on the previously visualized arterial tree. For the stage of matching limb severity, assessed through WIfI classification (a Global Limb Anatomic Staging System (GLASS) score obtained through DUAM, vein availability, patient’s periprocedural risk, and 2-year life expectancy), the best treatment strategy was planned, according to the evidence-based revascularization (EBR) [4].

All patients who offered the treatment were informed about revascularization and medical strategies selected by D.A. An informed consent was obtained from all patients who were undergoing revascularization and were included in the study.

Whenever needed, a surgical debridement of the foot was performed by diabetic foot specialists. It was planned before revascularization in the case of acute infection, and after revascularization when the foot was not infected, according to international guidelines for the management of DFU [17].

### 2.6. Follow-Up Protocol

Follow-up protocol included physical examination and DUS at discharge, 1, 3, 6, 12 months, and yearly thereafter for all patients. In selected cases, with a high suspicion of bypass stenosis and occlusion or with an inadequate DUS evaluation, patients underwent a second-level imaging to guide the further management.

### 2.7. Outcome Measures and Statistical Analysis

Primary outcomes were technical success and 30-day mortality. Secondary outcomes assessed were 1-year amputation free survival, need for second-level imaging, and time between evaluation and revascularization. Need for second-level imaging and time between evaluation and revascularization were analyzed and compared with data obtained in 2018, before the implementation of the diagnostic protocol.

All preoperative demographic features and examination details were recorded in a prospective database.

Continuous data were expressed as mean ± standard deviation (SD). Categorical data were expressed as percentages. The nonparametric Pearson chi-squared test was used to compare variables. The *t* Student test was used to compare two groups’ means. One-year amputation free survival was estimated using the Kaplan–Meier test.

A *p* value of <0.05 was considered to be statistically significant.

Statistical analysis was performed using SAS software version 9.4 (SAS Institute, Cary, NC, USA).

## 3. Results

### 3.1. Preoperative Demographic Data

Among the 128 patients initially evaluated, 68 (53.1%) underwent infrainguinal revascularization. A surgical revascularization consisted of femoropopliteal below-the-knee bypass (BTK) with great saphenous vein, in bifurcated popliteal–plantar BTK bypass, and in femoropopliteal BTK bypass with prosthetic graft. Then, endovascular revascularization implied plain balloon angioplasty (PBA) and drug-coated balloon angioplasty (DCB). Eventually, hybrid revascularization foresaw common femoral endoarterectomy alongside GSV femoropopliteal bypass and iliac stenting. The mean age was 73.6 ± 8.5 years. The majority of patients (50, 73.5%) were male and affected by type 2 diabetes mellitus (92.6%). An active smoke habit was present in 36.7% of cases. All patients suffered from mild to severe CLTI with a mean WIfI and GLASS scores of 5.2 ± 1.4 and 2.4 ± 0.7, respectively. Forty-two patients (61.7%) had a previous surgical or endovascular revascularization. Mean body mass index (BMI) was 27.04 ± 3.72.

Preoperative demographic features and comorbidities are summarized in Table 1.

### 3.2. Preoperative Results

DUS examination combined with clinical assessment to calculate the GLASS stage, suggested a surgical revascularization in 33 patients (48.5%), an endovascular treatment in 10 patients (14.7%), and an indeterminate area (‘grey zone’) in 25 patients (36.8%). Ultrasonographic criteria used to decide the best treatment among surgical, endovascular, or hybrid were multiple. Firstly, average-risk patients were candidate to BTK bypass if at least a valid runoff vessel was present, in case of superficial femoral artery (SFA) occlusion or stenosis > 25 cm in length or >75% in diameter, respectively, and in case of good-quality great saphenous vein (GSV) ≥ 3 mm. Secondly, patients received endovascular revascularization in case of prevalent infragenicular disease with at least two runoff vessels, and with infragenicular stenosis in any of the three vessels < 50%. An endovascular approach was also offered to patients with SFA stenosis < 25 cm in length and <75% in diameter. Eventually, a hybrid revascularization was performed in case of simultaneous inflow and outflow disease, such as iliac or common femoral artery (CFA) stenosis > 50% combined with one of the aforementioned infrainguinal or infragenicular features.

At the beginning and at the end of any procedure, be it surgical, endovascular, or hybrid, a DSA was performed in order to assess the correspondence between second-level imaging and DUS findings. Correspondence between DUS and DSA was complete for surgical treatment. Only 10 patients were judged suitable for endovascular treatment and a not negligible part was considered indeterminate with DUS. However, intraoperative angiography demonstrated a total correspondence between DUS and DSA even for patients treated using endovascular technique.

Hence, to confirm uncertain DUS results, a second-level imaging was necessary in 20 patients (29.4%), with a statistically significant difference compared to the number of second-level diagnostic examinations required for the 55 cases managed in the same outpatient setting in the year 2018 (*p* = 0.019). In addition, the time between evaluation and revascularization significantly decreased between 2018 and the study period (2018: 46 ± 8 days; 2022: 29 ± 17 days; *p* = 0.001).

Preoperative outcomes are summarized in Table 2.

### 3.3. Overall Outcomes

According to the GLASS score, a surgical revascularization was performed in 32 patients (47.0%). Twenty-four of these (75.0%) received a femoropopliteal below-the-knee bypass (BTK) with great saphenous vein, five (15.6%) received a bifurcated popliteal–plantar BTK bypass, and the remaining three (9.4%) underwent a femoropopliteal BTK bypass with prosthetic graft. Then, 34 patients (50.0%) received an endovascular revascularization, of which 29 (85.6%) with plain balloon angioplasty (PBA) and the remaining 5 (14.4%) with drug-coated balloon angioplasty (DCB). Eventually, two patients (3%) underwent a hybrid revascularization, combining common femoral endoarterectomy, GSV femoropopliteal bypass and iliac stenting. Overall technical success was achieved in 100% of the cases. The overall 30-day mortality rate was 2.9%. Only one patient underwent reintervention within the first 30 days for acute limb ischemia due to bypass occlusion. The mean follow-up was 11 ± 3 months. During the follow-up period, two patients (2.9%) died of cardiovascular complications. Ten patients (14.7%) had a recurrence for either bypass stenosis or occlusion. Specifically, six of them were treated with PB angioplasty to gain secondary patency, and the remaining four underwent surgical bypass extension. Complete wound healing was achieved in 40 patients (59.0%) with a mean healing time of 154.7 ± 120.9 days. One-year amputation free survival was 97.1%.

## 4. Discussion

The sharp development of new technologies and devices in the last decade promoted the growth of minimally invasive techniques in both treatment and diagnostic approaches. According to these progresses, DSA has frequently been replaced by less invasive methods, as MRI, CTA, and even more DUS. The latter provides several undisputable advantages compared to DSA. First, DSA means well-recognized risks of systemic and local complications. Systemic complications are mainly related to the potential nephrotoxicity of iodinated contrast agents, whereas local ones are mainly due to the arterial puncture [18,19,20,21,22]. Furthermore, DSA implies a non-negligible amount of ionizing radiation both for patients and operators, so that it cannot be frequently repeated. Moreover, several studies demonstrated that DUS and contrast-enhanced magnetic resonance angiography (CE-MRA) seemed to be superior to DSA in the prediction of the distal outflow vessels, especially those owing a very low blood flow [23,24].

Above all, DUS is absolutely risk-free, well-tolerated by the majority of patients, available in almost all settings, and is easy to perform at the first patient’s evaluation, compared to other diagnostic imaging. Then, the possibility to visualize both the lumen and the wall of target vessels is one of the main advantages of DUS, allowing us to characterize plaques in B-mode view. In addition, color-flow and power Doppler are essential tools to perform a hemodynamic evaluation, even in the case of poor distal runoff, where DSA cannot guarantee an optimal accuracy [25]. Obviously, a lack of patient’s compliance, obesity, as well as scars and wounds may be limitations to achieve an accurate DUAM. Above all, severe tibial vessel calcification is the most common cause of an incomplete DUS and determines when alternative imaging modalities need to be obtained [26]. Additionally, among the disadvantages of DUS, it would be well-advised to consider its duration and interoperator variability. Indeed, since DUS is used as a complete preoperative evaluation to plan revascularization, it can definitely be a time-consuming procedure. In order to reduce evaluation time, a short DUAM protocol combined with intraoperative graft pressure measurements has been proposed. However, the short protocol did not avert the need for completion arteriography and was tested only in claudicant patients [27]. Furthermore, whether DUS is performed by different operators without following standardized criteria, the discordance of results would be higher. Although the learning curve to achieve a high-quality DUAM may be long, especially for young surgeons, the standardized protocol used and taught in our unit allows us to overcome this issue.

Undoubtedly, DUAM represents a diagnostic noninvasive tool to easily assess vascular status in CLTI patients; however, it is not yet accepted as the main preoperative imaging method to plan revascularization, due to the aforementioned limitations. In this context, a review by Collins et al. [10] showed the inferiority of DUS to both CE-MRA and CTA in detecting some significant stenoses. This may constitute a main concern whenever DUS is used to screen patients before surgical procedures. However, DUS is unlikely to classify a lower limb as completely “free from any vascular disease” and therefore turndown patients from further diagnostic investigations. Concerning the comparison between DUS and intra-arterial DSA, Favaretto et al. [28] demonstrated poor agreement in the infrapopliteal districts, with a lower sensitivity of DUS in detecting significant stenoses or occlusions. However, more recent studies with an inverted trend have been published over the last several years, opening the possibility to use ultrasonography as a sole diagnostic method to plan revascularizations. Sensier et al. [29] examined a total of 51 limbs using both DUS and DSA, concluding that, since agreement between color duplex scanning and angiography never fell significantly below levels achieved between two different radiologists performing the investigations, color duplex ultrasound could be used to assess infrapopliteal artery patency. A following comparative study with a larger cohort confirmed these findings, showing that DUS correctly evaluated the status of runoff in 90% of the cases, with no significant differences in 30-day occlusion rate and patency at 12 months between revascularizations performed with and without preoperative angiography [30]. Despite that, these promising results have been challenged by the retrospective nature of all the studies mentioned. On the other hand, the prospective nature of our study allowed to avoid possible selection bias, guaranteeing more effective analyses. Interestingly, in our study, 36.8% of patients were considered indeterminate through DUS examination, so they were treated either surgically or endovascularly. However, intraoperative DSA demonstrated almost complete correspondence with DUS findings. Moreover, the extremely good intraoperative technical success rate witnessed the accuracy of this diagnostic tool in the preoperative assessment, precisely identifying the location and degree of stenosis and occlusions.

Many other comparisons have been developed during the years to demonstrate the superiority of DUS over other common imaging modalities apart from DSA. Recent comparative studies focused mainly on CTA, which remains the mainly used method to evaluate atherosclerotic lower limb occlusive disease and plan interventions. Despite its widespread use, this imaging modality shares some important limitations with DSA, since it always requires radiation and the use of intravenous iodine contrast medium to accurately opacify the arterial lumen. A recent analysis showed that, because of its accuracy in stenosis localization and hemodynamic evaluation, especially in below-the-knee segments, high-quality DUS may represent a good alternative to CTA overall [31]. These findings are completely in accordance with our results, where the concordance between DUS and second-level imaging was almost complete. Moreover, the extremely high success rate of our procedures with DUAM as a sole preoperative diagnostic tool, confirms not only the diagnostic accuracy of this method in guiding indications for treatment, but also its effectiveness in directing the appropriate choice among different revascularization strategies. In fact, in patients who underwent below-the-knee (BTK) bypass reconstructions, the site of the distal anastomoses planned through DUS were completely comparable with intraoperative angiographies. However, the reliability and accuracy of DUS can be limited for people living with Mönckeberg’s sclerosis secondary to diabetes due to arterial calcification, increasing difficulties to soundproof target vessels. In addition, it is reasonable to suggest that diabetic neuropathy may play a role and to hypothesize that neuropathic vasoconstriction may affect the waveform parameters. A novel parameter of Doppler waveforms, called the maximum systolic acceleration (AccMax) has been recently proposed as new accurate tool to overcome these issues. It represents the instantaneous systolic acceleration onset, not the time taken for blood to reach the PSV; therefore, it is plausible that AccMax is a more focused scope of measurement [32].

Furthermore, we demonstrated a 30-day mortality rate comparable with that reported in the literature [33], underlining the possibility to achieve equal outcomes regardless the imaging modalities used to plan the intervention. Another key point to take into account is that elderly CTLI patients or with multiple comorbidities, such as renal impairment, are those who mainly benefit from revascularization and non-invasive diagnostic assessments. Patients with severe renal impairment present the worse arteriosclerotic pattern of disease, and contrast medium infusion may cause further kidney deterioration. Although usually self-limited, contrast-induced acute kidney injury (AKI) may occasionally require hemodialysis, especially in patients with diabetes. This suggests that a comprehensive approach, including the preoperative diagnostic assessment, should be taken into consideration, especially when dealing with certain groups of patients. Therefore, the possibility to completely avoid or minimize the use of contrast medium in both diagnostic and planning phases, is definitely another advantage of DUS. Furthermore, it offers the possibility to evaluate the adequacy of inflow from iliac arteries to categorize the extension and the degree of the arterial lesions and eventually to guide the choice among outflow sites with regard to the runoff status. Obviously, not all patients are suitable for a single-method evaluation with only DUS. In fact, a non-negligible part of patients in our cohort required a second imaging modality because of uncertain DUS findings. When using ultrasonography, being aware of its intrinsic limitations, especially in the iliac axis and crural arteries, is mandatory. As already mentioned, heavily calcified arteries in diabetic patients or in those with a severe renal impairment can be challenging to visualize in their entire length. Similarly, collateral circles can be easily misjudged as patent arteries, leading to the choice to perform a further DSA before any interventions. However, the percentage of second-level imaging performed preoperatively in our unit has dramatically decreased over the years, since the “one step” multidisciplinary approach has been settled and the technical expertise in DUAM with a standardized protocol has improved. This fact seems to suggest that the frequency of second-level imaging and of inconclusive DUS may be further reduced with an increased expertise in ultrasonography. Moreover, the late presentation and delayed management of CLTI patients contribute to an increase in lower limb amputation and mortality rates [34]. Therefore, in our experience, a simultaneous evaluation performed by a multidisciplinary team, without the need of further imaging, allowed us to significantly reduce the time between evaluation and revascularization.

Surely, the main limitation of the present study is the small sample size that does not allow for the generalization of our results among the overall population. Secondly, as our data represent the experience of one group of surgeons at a single tertiary referral center for DFU, the probability of selection and information bias remains high. As an example of this kind of bias, the majority of patients participating in the study were recurrent ones, with at least one previous revascularization already performed in the index limb, mostly referred by other hospitals. Nevertheless, we feel that the information provided are relevant as a proof of concept for the indication of DUS as a sole diagnostic pre-operative approach for DFU patients with CLTI. Further multicenter prospective trials could confirm this hypothesis, which, if implemented in the guidelines, could save time and resources for the diagnostic work-up of these complex cases.

## 5. Conclusions

In conclusion, DUS can accurately evaluate the status of the entire vascular tree and the runoff on the foot, but also provides enough information about the target vessels to guide revascularizations’ strategy. In addition, it seems to have the potential role to replace common second-level diagnostic imaging, if performed by skilled operators in the framework of a multidisciplinary evaluation with the possibility to guarantee high technical success rates and satisfying short- and mid-term outcomes. However, the high operator-dependence of DUS compared to DSA is still limiting its widespread use as a sole preoperative imaging modality for planning a revascularization strategy in CLTI patients.

## Figures and Tables

**Figure 1 diagnostics-13-02879-f001:**
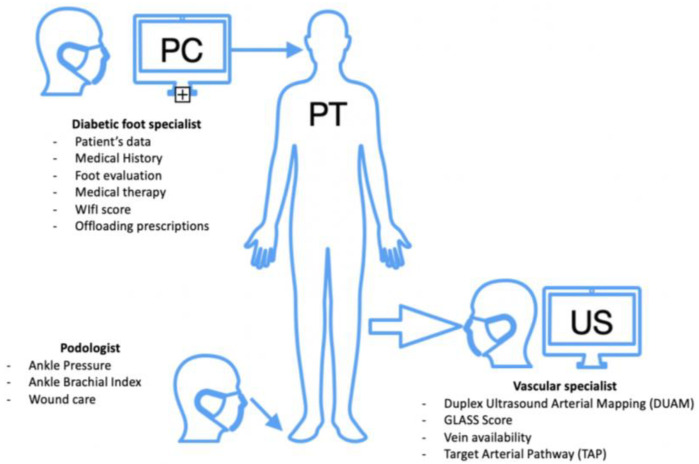
The picture illustrates our protocol of patients’ multidisciplinary evaluation.

**Table 1 diagnostics-13-02879-t001:** Summary of demographic features of the population study.

Demographic Features	Total of Patients (%)
Male sex	50 (73.5%)
Mean age ± SD	73.6 ± 8.5
Mean BMI ± SD	27.04 ± 3.72
Active smoking	25 (36.7%)
Comorbidities: Arterial hypertension Coronary artery disease Diabetes mellitus type I Diabetes mellitus type II Hypercholesterolemia Atrial fibrillation Cerebrovascular disease	63 (92.6%) 28 (41.2%) 5 (7.3%) 63 (92.6%) 30 (44.1%) 16 (23.5%) 7 (10.3%)
Previous revascularizations: Endovascular Surgical	36 (52.9%) 6 (8.8%)

**Table 2 diagnostics-13-02879-t002:** Summary of results after clinical and DUS evaluations.

Preoperative Results	Total of Patients (%)	*p* Value
Indication for revascularization (range 0–4) ± SD	3.6 ± 0.9	-
Type of revascularization suggested (GLASS):		
Surgical Endovascular Indeterminate	33 (48.5%) 10 (14.7%) 25 (36.8%)	-
Second-level imaging needed:		
2018 2022	49/55 (89.1%) 20/68 (29.4%)	0.019
Time between evaluation and revascularization:		
2018 2022	46 ± 8 days 29 ± 17 days	0.001

## Data Availability

Not applicable.

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
