# Peer review of "Peripheral Interventional Strategy Assessment (PISA) for Diabetic Foot Ulcer Revascularization: Preliminary Outcomes of a Multidisciplinary Pilot Study"

_diagnostics, 2023, doi:10.3390/diagnostics13182879_

Round 1

Reviewer 1 Report

Comments and Suggestions for Authors

Dear Authors,

Congratulations on your article "Peripheral Interventional Strategy Assessment (PISA) for diabetic foot ulcer revascularization: preliminary outcomes of a multidisciplinary pilot study". This article should be accepted under minor corrections.

Introduction

This chapter must be reduced and also more specific, approaching more directly the objectives of the paper.

Methods

The authors state that all the patients were submitted to a doppler ultrasound by the same vascular surgeon who had high experience in both doppler ultrasound and lower limb revascularization, however it is important to indicate other experience aspects, as: years in the active, number of exams conducted, among others.

Results

In line 179, the authors state patients underwent infra-inguinal revascularization. What kind of infra-inguinal revascularization was performed? This should be explained in more detail.

The overall outcomes indication must be improved (line 204), by including a more detailed description concerning: The type of intervention, either surgical or endovascular; Follow-up and its timing or if losses were registered and cicatrization.

Discussion

This chapter must also be shortened, by including less theoretical considerations. Instead a correlation should be made between theoretical considerations and results obtained. Disadvantages of the Eco-Doppler must be stated, including time consuming intervention, differential technique, among others, additionally the advantages related to the fact that it can be performed at the moment. The authors should indicate if in the patients submitted to endovascular intervention there was a match between the images of the doppler ultrasound results and geography.

In the methods or results the authors should make clear if the surgeon who performed the doppler ultrasound was also the same of the surgeries in all the patients.

Best regards, 

Joana Ferreira

Comments on the Quality of English Language

Only moderate editing of english is necessary.

Reviewer 2 Report

Comments and Suggestions for Authors

Well-written study examining the use of Duplex exclusively as a means to guide revascularization.  I have the following questions/comments:

1. Line 112-113. Given the experience of the surgeon with Duplex, can the authors comment on how this can translate to other practices and the learning curve involved?

2. Line 134-7.  How often was CTA or angiography employed and what % were for aortoiliac disease, fem-pop disease, or tibial/crural disease?

3. Line 184.  Would include the BMI of the patients.

4. Line 183-4.  61.7% had prior revascularization.  Can the authors comment on how that can potentially bias the Duplex - there is already going to be reasonable knowledge of the existing inflow and outflow targets in these cases.

5. Table 1.  The Wifi, Glass, and Rutherford scores are not labeled and out of context.  The labeling suggests this is the % of patients who were scored in each category.

6. Line 204-9. What was the % of distal popliteal target vs. tibial or pedal target?

Reviewer 3 Report

Comments and Suggestions for Authors

1.     Introduction, page 2, line 51: The VASCUQOL is generally accepted as a disease specific QOL assessment for patients with CLTI.

2.     Methods, page 4, line 132: I do not know what the authors mean by vein heritage.

3.     Results, page 5, line 179: What was the treatment and outcomes of the 46.9% of patients that did not get revascularization and why was this chosen? That seems like a low rate of revascularization for a group of patients with CLTI.

4.     Results, page 6, line 191: What were the ultrasound criteria used to determine endovascular vs open vs hybrid revascularization?

5.     Results: Please comment on the correlation between angiographic and duplex findings on patients who underwent endovascular revascularization.

6.     Results: For surgical revascularization, was any intraoperative imaging performed to confirm duplex findings?

7.     Discussion: Theresa Jacob and Enrico Ascher have published extensively on duplex ultrasound arterial mapping. The authors should include some of their papers in the discussion and references.

8.     Discussion, page 8, line 297: For BTK lesions, patients with diabetes often have very calcified vessels, making insonation of these vessels very difficult. Could the authors discuss techniques and limitations of doing duplex arterial mapping in these patients?

Comments on the Quality of English Language

The paper needs extensive English language editing. In particular, the introduction and discussion are too long and not well organized, making them very difficult to read. 

Round 2

Reviewer 3 Report

Comments and Suggestions for Authors

no additional questions

Comments on the Quality of English Language

paper is improved but still requires some English language editing